Risk factors for bronchopulmonary dysplasia in preterm infants: a systematic review and meta-analysis

Xiong Ping 1
Li Lei 1
Yu Zhangbin 2
Pu Yuanlin 1 214213990@qq.com
Tang Hong 3 superyict@126.com
1 Department of Neonatology, The Central Hospital of Enshi Tujia and Miao Autonomous Prefecture , Enshi, Hubei , China
2 Department of Neonatology, Shenzhen People’s Hospital (The Second Clinical Medical College, Jinan University; The First Affiliated Hospital, Southern University of Science and Technology) , Shenzhen, Guangdong , China
3 Department of Neonatology, Shenzhen Yantian District People’s Hospital , Shenzhen, Guangdong , China
Anson Lesley
Electronic publication date: 2025 Oct 10
Publication date: 2025
Volume: 13
Electronic Location ID: e20202
Received 2025 Jan 13; Accepted 2025 Sep 16
Copyright: © 2025 Xiong et al.
Copyright year: 2025
Copyright holder: Xiong et al.
License: This is an open access article distributed under the terms of the Creative Commons Attribution License, which permits unrestricted use, distribution, reproduction and adaptation in any medium and for any purpose provided that it is properly attributed. For attribution, the original author(s), title, publication source (PeerJ) and either DOI or URL of the article must be cited.
License URL: https://creativecommons.org/licenses/by/4.0/

Keywords: Bronchopulmonary dysplasia, Preterm infants, Risk factors, Meta-analysis

Funding: The authors received no funding for this work.

==============================
Background

Bronchopulmonary dysplasia (BPD) is the most common respiratory disease in preterm infants. As medical advancements have increased the survival rate of preterm infants, the prevalence of BPD has also increased, representing a significant societal burden. The pathogenesis of BPD is multifactorial, involving both genetic and environmental factors. Although numerous studies have examined risk factors for BPD, their findings are inconsistent. Few meta-analyses exist, yet most focus on risk factors for the development of pulmonary hypertension in infants with BPD. The primary aim of this study was to identify the risk factors for BPD.

Methods

The study protocol was registered with PROSPERO (CRD42024616871). A comprehensive literature search was conducted in the PubMed, Embase, Cochrane Library, and Web of Science databases for case-control and cohort studies investigating risk factors for BPD. The search was completed on 22 November 2024, and the data were analyzed using Review Manager 5.3.5 and Stata 15.1.

Results

A total of 23 studies were included in the analysis, encompassing 14,729 patients in the bronchopulmonary dysplasia (BPD) group and 19,101 in the non-bronchopulmonary dysplasia (non-BPD) group. The meta-analysis revealed that chorioamnionitis (CA) was associated with an increased risk of BPD (OR = 1.52, 95% CI [1.23–1.87]), as was premature rupture of membranes (PROM; OR = 1.42, 95% CI [1.02–1.98]). Additionally, hypertensive disorders of pregnancy (HDP) were identified as a significant risk factor for BPD (OR = 2.73, 95% CI [1.31–5.69]). Other notable risk factors included lower gestational age (GA; MD = −1.86, 95% CI [−2.35 to −1.38]), male sex (OR = 1.41, 95% CI [1.14–1.75]), and being small for gestational age (SGA; OR = 3.14, 95% CI [1.03–9.60]). Furthermore, the analysis indicated that mechanical ventilation (MV; MD = 16.55, 95% CI [9.68–23.41]), oxygen administration (MD = 50.91, 95% CI [37.40–64.42]), and blood transfusion (OR = 1.38, 95% CI [1.06–1.81]) were significant risk factors for BPD. Other variables that were identified as significant risk factors included patent ductus arteriosus (PDA; OR = 1.75, 95% CI [1.35–2.27]), sepsis (OR = 1.88, 95% CI [1.44–2.46]), and respiratory distress syndrome (RDS; OR = 6.37, 95% CI [4.00–10.13]).

Conclusions

Significant risk factors for BPD include CA, PROM, HDP, lower GA, male sex, SGA, MV, oxygen administration, blood transfusions, PDA, sepsis, and RDS. These findings hold potential clinical significance for predicting BPD pathogenesis.

Introduction

A systematic analysis based on data from 103 countries and territories showed that the global prevalence of preterm birth was 9.9% in 2020, with approximately 15 million preterm births per year (Ohuma et al., 2023). Bronchopulmonary dysplasia (BPD) remains the most common complication associated with prematurity and is increasing in prevalence (Thébaud et al., 2019). A Chinese study showed that the incidence of BPD was 29.2% in preterm infants with a gestational age of less than 32 weeks and as high as 47.8% in those born before 28 weeks (Cao et al., 2021).

BPD is a chronic lung disease in preterm infants, defined by a persistent dependence on supplemental oxygen and/or respiratory support beyond 28 days of life or at 36 weeks postmenstrual age (PMA) (Higgins et al., 2018; Jensen et al., 2019; Jobe & Bancalari, 2001). It can be attributable to preterm birth, mechanical ventilation, oxygen toxicity, infection, and genetic factors (Dankhara et al., 2023). Preterm infants with BPD have a longer initial hospital stay than those without BPD (Thébaud et al., 2019). After neonatal intensive care unit discharge, infants with BPD often require frequent hospital readmissions and have high rates of emergency department and physician visits due to recurrent respiratory exacerbations, lower respiratory tract infections, reactive airway disease, and pulmonary hypertension (Taglauer, Abman & Keller, 2018; Thébaud et al., 2019). There is currently no specific treatment available for BPD, and the main clinical focus for BPD is based on symptomatic treatment (Chang, 2018). Therefore, it is crucial to identify risk factors for BPD to help actively prevent the disease.

The primary objective of this study was to identify risk factors for BPD across three critical perinatal periods (antenatal, intrapartum, and postnatal), thereby providing a foundation for clinical decision-making and enabling the early prediction and prevention of this condition.

Method

The primary aim of this study was to identify the main factors leading to BPD during the antenatal, intrapartum, and postnatal periods, and to lay the groundwork for early preventive measures. This study was registered in the PROSPERO International Prospective Register of Systematic Reviews (Registration number: CRD42024616871) and was conducted following the PRISMA guidelines (Liberati et al., 2009).

Search strategy

A comprehensive search was conducted in the PubMed, Embase, Cochrane Library, and Web of Science databases from inception through 22 November 2024. The search strategy used a combination of Medical Subject Headings (MeSH) terms and free text terms including “neonate,” “preterm,” “bronchopulmonary dysplasia,” and “risk factors”. Manual searches of references from relevant articles and reviews were conducted to ensure comprehensive coverage. A detailed account of the search strategy is provided in Table S1 (Appendix 1).

Definitions of key terms

Chorioamnionitis (CA) was diagnosed by clinical findings including maternal fever, leukocytosis, and uterine tenderness, or the presence of acute inflammatory changes observed in the tissue samples collected from the amnion, chorion, and parietal decidua (Mittendorf et al., 2005). Mechanical ventilation (MV) was restricted to invasive ventilation via endotracheal tube (non-invasive modes excluded). Oxygen administration was defined as the administration of oxygen at concentrations greater than 21%, including delivery methods such as oxygen hoods, nasal cannulas, and face masks. Patent ductus arteriosus (PDA) was diagnosed based on signs of a murmur, bounding pulses, active precordium, and echocardiography (Dou et al., 2023). Treated PDA was defined as PDA requiring medical or surgical treatment based on both echocardiographic and clinical findings (Nakashima et al., 2021). Neonatal sepsis was classified into early-onset sepsis (EOS, occurring within 72 h after birth) and late-onset sepsis (LOS, occurring after 72 h postpartum). Diagnosis types included clinically diagnosed sepsis (meeting clinical criteria for sepsis but with negative blood cultures) and confirmed sepsis (blood culture-positive) (Abushahin et al., 2024).

Inclusion and exclusion criteria

Inclusion criteria: 1. The target population for the study comprises preterm infants born at a gestational age ≤32 weeks and/or with a birth weight ≤1,500 g.

2. The study design includes case-control and/or cohort studies.

3. The focus of the study is on risk factors or influencing factors for BPD (Bronchopulmonary dysplasia [BPD] was defined based on the National Institute of Child Health and Human Development (NICHD) consensus criteria (2001) as oxygen dependence for at least 28 days, with severity classified by respiratory support requirements at 36 weeks postmenstrual age (Jobe & Bancalari, 2001)).

4. Clear outcome measures are available from the original articles, including, but not limited to, computable outcome measures such as odds ratios (OR) and 95% confidence intervals (CI).

5. The most comprehensive or recent publication is selected when articles have similar authors and content.

6. The article is published in English language.

Exclusion criteria: 1. BPD is not diagnosed according to the 2001 National Institutes of Child Health and Human Development (NICHD) criteria.

2. In the original study design, death and/or BPD are the control group or outcome measure.

3. Conference abstracts, meta-analyses, protocols, letters, systematic reviews, and animal studies.

4. Non-English-language publications.

5. Studies for which the full text is not available.

6. Studies with duplicate data.

7. Studies with incomplete data.

Study selection and data extraction

Two researchers (PX and LL) conducted independent literature searches and data extraction. During the screening process, the established inclusion and exclusion criteria were rigorously adhered to. In instances where there was a divergence of opinion, the input of experts was sought (YP and HT). The relevant indicators from the studies were extracted and cross-checked to ensure consistency. The primary data extraction content included the first author, year of publication, country of origin, study period, type of study, study population, sample size, and risk factors. The risk factors were primarily derived from the results of multifactor analyses and some combined data.

Quality assessment

As the included studies were case-control or cohort studies, the Newcastle-Ottawa Scale (NOS) was selected for quality assessment. The evaluation encompassed eight questions distributed across three domains: study population, comparability between groups, and outcome measures/exposure factor measures. The first domain comprised four questions, the second domain comprised one question, and the third domain comprised three questions. Each of the seven questions was awarded one point with the exception of comparability, which was assigned two points (Stang, 2010). The total possible score was nine points; a score of three or less indicated low quality, a score of four to six indicated medium quality, and a score of seven or above indicated high quality (Lo, Mertz & Loeb, 2014). In the event of a discrepancy between the two researchers during the evaluation process, the decision was discussed with all authors.

Statistical analysis

The statistical analysis was conducted using Review Manager 5.3.5 and Stata 15.1. Risk values from studies could be expressed as an OR or as a mean difference (MD) with a 95% CI. These values were considered the effect size, and effect sizes were pooled for the same factor mentioned in two or more original studies. Heterogeneity was evaluated using I2 statistics and Cochran’s Q test, with studies considered heterogeneous (I2 > 50% or Q-test P ≤ 0.1) or homogeneous (I2 ≤ 50% and Q-test P > 0.1). Accordingly, a random-effects model was used when heterogeneity was present, while a fixed-effects model was applied when studies were homogeneous for calculating the pooled OR/MD. The significance level was set at α = 0.05, with P < 0.05 deemed statistically significant. In cases where a minimum of 10 original studies were available for a specific factor, funnel plots were combined with an Egger’s test to assess the potential for publication bias.

Results

Literature search results

A preliminary search of the PubMed, Embase, Cochrane Library, and Web of Science databases yielded 6,700 articles. After deduplication, 4,794 articles remained. The titles and abstracts of these articles were then reviewed, resulting in the selection of 239 articles. Following a full-text review, 23 articles were included in the final analysis (Abushahin et al., 2024; Alonso et al., 2022; Benali et al., 2024; Bolat, Dursun & Sarlaydln, 2024; Cokyama & Kavuncuoglu, 2020; Demirel, Bas & Zenciroglu, 2009; Dou et al., 2023; Duan et al., 2016; Ebrahimi et al., 2021; Gobec et al., 2023; Guimarães et al., 2010; Huang et al., 2023; Jung & Lee, 2019; Nakashima et al., 2021; Ozkan, Cetinkaya & Koksal, 2012; Park, Bae & Chang, 2021; Patel et al., 2019; Rocha et al., 2019a, 2019b; Rojas et al., 2012; Rutkowska et al., 2019; Shin et al., 2020; Zhang, Huang & Lu, 2014). The specific search process is illustrated in Fig. 1.

Figure 1 Flowchart of literature search.

A systematic search of PubMed, Embase, Cochrane Library, and Web of Science identified 6,700 records. After removing 1,906 duplicates, 4,794 records were screened by title/abstract. Of these, 4,555 were excluded, leaving 239 full-text articles assessed for eligibility. Finally, 23 studies met inclusion criteria for meta-analysis.

Characteristics of the included studies

A total of 23 studies were included in the review. Multiple studies originated from Turkey (three studies), Korea (three studies), Portugal (three studies), and China (four studies), while single studies were identified from the United States, Qatar, Tunisia, Slovenia, Spain, Holland, Japan, Poland, Colombia, and India. The BPD group comprised 14,729 patients and the non-BPD group comprised 19,101 individuals. The characteristics of the included studies are detailed in Table 1.

Table 1 Characteristics of included studies.

The table summarizes key features of the studies included in the analysis, such as reference, location, duration, study population, and risk factors.

References	Location	Duration	Study population	Size	Risk factors	NOS	
Non-BPD	BPD	
Case-control study	
Benali et al. (2024)	Tunisia	2017–2021	GA 26–31 w	144	87	(1) (3) (7) (8)	5	
Dou et al. (2023)	China	2018–2021	GA 23–31 w	1,646	823	(7) (10) (11)	7	
Cokyama & Kavuncuoglu (2020)	Turkey	2006–2008	GA ≤ 32 w	551	139	(2) (4) (5) (10) (11) (12)	6	
Rojas et al. (2012)	Colombia	2004–2006	GA 27–31 w	148	64	(2) (4) (10) (11)	6	
Demirel, Bas & Zenciroglu (2009)	India	2004–2006	BW ≤ 1,500 g	50	56	(10) (12)	6	
Cohort study	
Abushahin et al. (2024)	Qatar	2017–2020	GA ≤ 32 w	1,088	451	(4) (5) (7) (10) (11)	7	
Bolat, Dursun & Sarlaydln (2024)	Turkey	2016–2010	GA < 32 w and/or BW < 1,500 g	139	107	(4) (7) (9)	7	
Gobec et al. (2023)	Slovenia	2009–2019	GA < 28 w	76	113	(4) (7) (8)	6	
Huang et al. (2023)	China	2016–2019	GA < 30 w or BW ≤ 1,500 g	1,153	1,958	(1) (4) (10) (11)	7	
Alonso et al. (2022)	Spain	2013–2020	GA ≤ 32 w and BW ≤ 1,500 g	144	58	(4) (11)	7	
Ebrahimi et al. (2021)	Holland	2009–2015	GA < 30 w	130	79	(4) (7)	6	
Nakashima et al. (2021)	Japan	2003–2016	GA < 32 w or BW ≤ 1,500 g	9,334	7,792	(1) (6) (10) (11)	6	
Park, Bae & Chang (2021)	Korea	2014–2017	GA 23–27 w	503	381	(1) (2)	6	
Shin et al. (2020)	Korea	2013–2014	GA < 30 w	1,063	764	(1) (3) (5) (6) (10)	7	
Patel et al. (2019)	USA	2010–2013	BW < 1,500 g	358	240	(9)	7	
Rocha et al. (2019a)	Portugal	2015–2016	GA 24–30 w	375	119	(6)	6	
Rutkowska et al. (2019)	Poland	2014–2015	GA ≤ 32 w	388	319	(4) (5)	6	
Rocha et al. (2019b)	Portugal	2015–2016	GA 24–30 w	375	119	(3) (4) (8)	6	
Jung & Lee (2019)	Korea	2013–2015	BW < 1,000 g	618	816	(4) (5) (11)	7	
Duan et al. (2016)	China	2014–2015	GA < 32 w	172	71	(9)	7	
Zhang, Huang & Lu (2014)	China	2008–2013	GA ≤ 32 w or BW ≤ 1,500 g	178	53	(9)	7	
Ozkan, Cetinkaya & Koksal (2012)	Turkey	2009–2010	GA ≤ 32 w	245	87	(3)	7	
Guimarães et al. (2010)	Portugal	2004–2006	GA < 30 w or BW < 1,250 g	223	33	(4) (7) (8) (11) (12)	7	
Note:

(1) Chorioamnionitis (CA); (2) Premature rupture of membranes (PROM); (3) Hypertensive disorders of pregnancy (HDP); (4) Gestational age (GA); (5) Male; (6) Small for gestational age (SGA); (7) Mechanical ventilation (MV); (8) Oxygen exposure; (9) Blood transfusion; (10) Patent ductus arteriosus (PDA); (11) Sepsis; (12) Respiratory distress syndrome (RDS); (13) Newcastle-Ottawa Scale (NOS).

Methodological quality and risk of bias

The Newcastle-Ottawa Scale (NOS) was used to evaluate the quality of the included studies, with one study awarded a rating of five, ten studies assigned a rating of six, and the remaining twelve studies receiving a rating of seven. This suggests that the overall quality of the studies included was high. The specific quality ratings are presented in Table 1 and Table S2 (Appendix 2).

Risk factors for BPD

Chorioamnionitis

The topic of chorioamnionitis (CA) was addressed in five studies (Benali et al., 2024; Huang et al., 2023; Nakashima et al., 2021; Park, Bae & Chang, 2021; Shin et al., 2020), with notable heterogeneity observed (I2 = 55% > 50%, P = 0.06 < 0.1). A random-effects model was used for the meta-analysis. The findings indicated that infants exposed to CA had significantly higher odds of developing BPD compared to unexposed infants (OR = 1.52, 95% CI [1.23–1.87]), with a statistically significant difference (P < 0.0001; Table 2; Appendix 3).

Table 2 Analysis of factors influencing BPD.

Presents pooled estimates of BPD-associated factors, including: sample size (Non-BPD/BPD groups), effect sizes (OR/MD with 95% CI), significance tests (Z/P), and heterogeneity (I2/P).

Factor	No of study	Sample size	Effect size	Effects model	Global effects test	Heterogeneity	
Non-BPD	BPD	OR (95% CI)	MD (95% CI)	Z	P	I2 (%)	P	
Antenatal risk factors	
CA	5	12,197	10,982	1.52 [1.23–1.87]		R	3.91	<0.0001	55	0.06	
PROM	3	1,202	584	1.42 [1.02–1.98]		R	2.06	0.04	63	0.07	
HDP	4	1,827	1,057	2.73 [1.31–5.69]		R	2.68	0.007	69	0.02	
Intrapartum risk factors	
GA	12	5,033	4,256		−1.86 [−2.35 to −1.38]	R	7.5	<0.00001	97	<0.00001	
Male	5	3,708	2,489	1.41 [1.14–1.75]		R	3.17	0.002	51	0.08	
SGA	3	10,772	8,675	3.14 [1.03–9.60]		R	2.01	0.04	86	0.0009	
Postnatal risk factors	
MV	7	3,446	1,693		16.55 [9.68–23.41]	R	4.72	<0.00001	98	<0.00001	
Oxygen
administration	4	818	352		50.91 [37.4–64.42]	R	7.39	<0.00001	97	<0.00001	
Blood transfusion	4	847	471	1.38 [1.06–1.81]		R	2.37	0.02	86	<0.0001	
PDA	8	15,033	12,047	1.75 [1.35–2.27]		R	4.22	<0.0001	91	<0.00001	
Sepsis	9	14,905	12,134	1.88 [1.44–2.46]		R	4.65	<0.00001	87	<0.00001	
RDS	3	824	228	6.37 [4.0–10.13]		F	7.82	<0.00001	0	0.93	
Note:

R, Random effects model; F, Fixed effects model; CA, Chorioamnionitis; PROM, Premature rupture of membranes; HDP, Hypertensive disorders of pregnancy; GA, Gestational age; SGA, Small for gestational age; MV, Mechanical ventilation; PDA, Patent ductus arteriosus; RDS, Respiratory distress syndrome.

Premature rupture of membranes

The topic of premature rupture of membranes (PROM) was discussed in three studies (Cokyama & Kavuncuoglu, 2020; Park, Bae & Chang, 2021; Rojas et al., 2012), with a notable degree of heterogeneity (I2 = 63% > 50%, P = 0.07 < 0.1). A random-effects model was used for the meta-analysis. The findings indicated that PROM was a risk factor for BPD (OR = 1.42, 95% CI [1.02–1.98]), with a statistically significant difference (P = 0.04 < 0.05; Table 2; Appendix 4).

Hypertensive disorders of pregnancy

The topic of hypertensive disorders of pregnancy (HDP) was addressed in four studies (Benali et al., 2024; Ozkan, Cetinkaya & Koksal, 2012; Rocha et al., 2019b; Shin et al., 2020). The meta-analysis used a random-effects model due to the presence of significant heterogeneity (I2 = 69% > 50%, P = 0.02 < 0.1). The analysis revealed that HDP was a risk factor for BPD (OR = 2.73, 95% CI [1.31–5.69]), with a statistically significant difference (P = 0.07 < 0.05; Table 2; Appendix 5).

Gestational age

The topic of gestational age (GA) was addressed in twelve studies (Abushahin et al., 2024; Alonso et al., 2022; Bolat, Dursun & Sarlaydln, 2024; Cokyama & Kavuncuoglu, 2020; Ebrahimi et al., 2021; Gobec et al., 2023; Guimarães et al., 2010; Huang et al., 2023; Jung & Lee, 2019; Rocha et al., 2019b; Rojas et al., 2012; Rutkowska et al., 2019), with significant heterogeneity observed (I2 = 97% > 50%, P < 0.1). A random-effects model was used for the meta-analysis. The findings showed that infants with BPD had a lower GA than those without BPD (MD = −1.86, 95% CI [−2.35 to −1.38]), with a statistically significant difference (P < 0.00001; Table 2; Appendix 6).

Sex

Sex was discussed in five studies (Abushahin et al., 2024; Cokyama & Kavuncuoglu, 2020; Jung & Lee, 2019; Rutkowska et al., 2019; Shin et al., 2020), with significant heterogeneity observed (I2 = 51% > 50%, P = 0.08 < 0.1). A random-effects model was used for the meta-analysis. The results suggested that male sex was a risk factor for BPD (OR = 1.41, 95% CI [1.14–1.75]), with a statistically significant difference (P = 0.002; Table 2; Appendix 7).

Small for gestational age

The topic of small for gestational age (SGA) was addressed in three studies (Nakashima et al., 2021; Rocha et al., 2019a; Shin et al., 2020), with notable heterogeneity observed (I2 = 86% > 50%, P < 0.1). A random-effects model was used for the meta-analysis. The findings indicated that SGA was a risk factor for BPD (OR = 3.14, 95% CI [1.03–9.60]), with a statistically significant difference (P = 0.04; Table 2; Appendix 8).

Mechanical ventilation

The topic of mechanical ventilation (MV) was addressed in seven studies (Abushahin et al., 2024; Benali et al., 2024; Bolat, Dursun & Sarlaydln, 2024; Dou et al., 2023; Ebrahimi et al., 2021; Gobec et al., 2023; Guimarães et al., 2010), with significant heterogeneity observed (I2 = 98% > 50%, P < 0.1). A random-effects model was used for the meta-analysis. A significantly extended duration of MV was observed in BPD infants vs. non-BPD controls (MD = 16.55, 95% CI [9.68–23.41]), with a statistically significant difference (P < 0.00001; Table 2; Appendix 9).

Oxygen administration

The topic of oxygen administration was addressed in four studies (Benali et al., 2024; Gobec et al., 2023; Guimarães et al., 2010; Rocha et al., 2019b), with notable heterogeneity observed (I2 = 97% > 50%, P < 0.1). A random-effects model was used for the meta-analysis. The results demonstrated that the duration of oxygen administration was significantly prolonged in the BPD group compared to the non-BPD group (MD = 50.91, 95% CI [37.40–64.42]), with a statistically significant difference (P < 0.00001; Table 2; Appendix 10).

Blood transfusion

The topic of blood transfusion was addressed in four studies (Bolat, Dursun & Sarlaydln, 2024; Duan et al., 2016; Patel et al., 2019; Zhang, Huang & Lu, 2014), with considerable heterogeneity observed (I2 = 86% > 50%, P < 0.1). A random-effects model was used for the meta-analysis. The results indicated that blood transfusion was a risk factor for BPD (OR = 1.38, 95% CI [1.06–1.81]), with a statistically significant difference (P = 0.02; Table 2; Appendix 11).

Patent ductus arteriosus

The topic of patent ductus arteriosus (PDA) was analyzed in eight studies (Abushahin et al., 2024; Cokyama & Kavuncuoglu, 2020; Demirel, Bas & Zenciroglu, 2009; Dou et al., 2023; Huang et al., 2023; Nakashima et al., 2021; Rojas et al., 2012; Shin et al., 2020), with notable heterogeneity observed (I2 = 91% > 50%, P < 0.1). A random-effects model was used for the meta-analysis. The results indicate that PDA was a risk factor for BPD (OR = 1.75, 95% CI [1.35–2.27]), with a statistically significant difference (P < 0.0001; Table 2; Appendix 12).

Sepsis

The topic of sepsis was addressed in nine studies (Abushahin et al., 2024; Alonso et al., 2022; Cokyama & Kavuncuoglu, 2020; Dou et al., 2023; Guimarães et al., 2010; Huang et al., 2023; Jung & Lee, 2019; Nakashima et al., 2021; Rojas et al., 2012), with notable heterogeneity observed (I2 = 87% > 50%, P < 0.1). A random-effects model was used for the meta-analysis. The results indicated that sepsis was a risk factor for BPD (OR = 1.88, 95% CI [1.44–2.46]), with a statistically significant difference (P < 0.00001; Table 2; Appendix 13).

Respiratory distress syndrome

Respiratory distress syndrome (RDS) was reported in three studies (Cokyama & Kavuncuoglu, 2020; Demirel, Bas & Zenciroglu, 2009; Guimarães et al., 2010) and was analyzed with a fixed effects model with no significant heterogeneity observed (I2 = 0%, P = 0.93). The analysis suggested that RDS was a significant risk factor for BPD (OR = 6.37, 95% CI [4.0–10.13]), with a statistically significant difference (P < 0.00001; Table 2; Appendix 14).

Subgroup analysis

This study investigated the association between gestational age (GA) and BPD across different study designs using subgroup analysis. The pooled results from the case-control subgroup (Cokyama & Kavuncuoglu, 2020; Rojas et al., 2012) showed no statistically significant difference in GA between the BPD and non-BPD groups (MD = −2.10 weeks, 95% CI [−4.84 to 0.65], P = 0.13), but extremely high heterogeneity (I2 = 99%) indicated limited reliability of the findings. In contrast, the cohort study subgroup (Abushahin et al., 2024; Alonso et al., 2022; Bolat, Dursun & Sarlaydln, 2024; Ebrahimi et al., 2021; Gobec et al., 2023; Guimarães et al., 2010; Huang et al., 2023; Jung & Lee, 2019; Rocha et al., 2019b; Rutkowska et al., 2019) demonstrated significantly lower GA in the BPD group compared to the non-BPD group (P < 0.00001), though substantial heterogeneity remained (I2 = 97%), warranting cautious interpretation. The test for subgroup differences revealed no statistically significant effect of study design on the GA-BPD association (P = 0.84, I2 = 0%). Overall, while cohort studies supported a significant link between lower GA and BPD, the high heterogeneity across both study types suggests the need for further investigation into potential confounding factors (Table 3 and Appendix 15).

Table 3 Subgroup analysis.

Presents subgroup analyses evaluating associations between BPD and gestational age, mechanical ventilation, PDA, and sepsis. Data include subgroup sample sizes, effect estimates (OR/MD with 95% CI), heterogeneity (I2), and statistical significance (P-values).

Factor	No of study	Sample size	Heterogeneity	Effects model	Effect size	Global effects test	
Non-BPD	BPD	I2 (%)	P	OR (95% CI)	MD (95% CI)	P	
GA (Type of study)	
Case-control study	2	699	203	99%	<0.00001	R		−2.10 [−4.84 to 0.65]	0.13	
Cohort study	10	1,202	584	97%	<0.00001	R		−1.82 [−2.31 to −1.33]	<0.00001	
MV (Gestational age)	
GA < 30 weeks	3	429	225	97%	<0.00001	R		21.28 [5.81–36.76]	0.007	
GA ≤ 32 weeks	4	3,017	1,468	99%	<0.00001	R		13.8 [4.82–22.79]	0.003	
Sepsis (Gestational age)	
GA < 30 weeks	3	1,994	2,807	67%	0.05	R	2.14 [1.33–3.42]		0.002	
GA ≤ 32 weeks	6	12,911	9,327	88%	<0.00001	R	1.76 [1.29–2.39]		0.0003	
Sepsis (Diagnosis)	
LOS	5	4,047	2,287	65%	0.02	R	2.25 [1.61–3.13]		<0.00001	
EOS and LOS	4	10,858	9,847	85%	0.0002	R	1.42 [1.06–1.89]		0.02	
PDA (Gestational age)	
GA < 30 weeks	2	2,216	2,722	92%	0.0003	R	1.72 [0.98–3.02]		0.06	
GA ≤ 32 weeks	6	12,817	9,325	89%	<0.00001	R	1.74 [1.29–2.35]		<0.00001	
PDA (No treatment or treatment)	
PDA	5	4,488	3,427	85%	<0.0001	R	2.31 [1.44–3.71]		0.0006	
PDA (Treated)	3	10,545	8,620	94%	<0.00001	R	1.40 [1.02–1.93]		0.04	
Note:

R, Random effects model; GA, Gestational age; MV, Mechanical ventilation; PDA, Patent ductus arteriosus; LOS, Late-onset sepsis; EOS, Early-onset sepsis.

This study investigated the impact of different GA stratifications on MV duration through subgroup analysis. In the very preterm infant subgroup (GA < 30 weeks), the pooled results demonstrated a significantly prolonged MV duration in BPD vs. non-BPD infants (MD = 21.28, 95% CI [5.81–36.76], P = 0.007), albeit with extremely high heterogeneity (I2 = 97%). Similarly, in the very preterm infant subgroup (GA ≤ 32 weeks), MV duration was also significantly extended in BPD vs. non-BPD infants (MD = 13.80, 95% CI [4.82–22.79], P = 0.003), with even higher heterogeneity (I2 = 99%). Notably, the test for subgroup differences showed no statistical significance (P = 0.41, I2 = 0%), indicating that GA stratification did not significantly affect the overall MV duration. These findings suggest that infants with GA ≤ 32 weeks, particularly GA < 30 weeks, may require longer mechanical ventilation support. However, the substantial heterogeneity among studies (I2 = 97–99%) implies potential variations in clinical practice or other confounding factors, warranting further investigation for clarification (Table 3 and Appendix 16).

This meta-analysis examined the association between sepsis and the risk of BPD in preterm infants, with subgroup analyses across different GAs. The results demonstrated that sepsis significantly increased the risk of BPD (pooled OR = 1.88, 95% CI [1.44–2.46], P < 0.00001), and this association remained significant across all GA subgroups. Notably, in preterm infants with sepsis exposure compared to those without, very preterm infants with GA < 30 weeks exhibited a higher risk (OR = 2.14, 95% CI [1.33–3.42]), while those with GA ≤ 32 weeks showed a slightly lower but still significant risk (OR = 1.76, 95% CI [1.29–2.39]). Tests for subgroup differences indicated no significant effect of GA stratification on the results (P = 0.50; Table 3 and Appendix 17), despite substantial heterogeneity (I2 = 67–87%), which may stem from variations in diagnostic classifications or management strategies. Subgroup analysis based on different diagnostic classifications of sepsis revealed that the LOS group had significantly higher risk (OR = 2.25, 95% CI [1.61–3.13]) compared to the mixed EOS/LOS group (OR = 1.42, 95% CI [1.06–1.89]), with a statistically significant intergroup difference (P = 0.04). Notably, both groups exhibited varying degrees of heterogeneity (LOS group I2 = 65%, mixed group I2 = 85%), potentially attributable to differences in diagnostic criteria or study design. The findings demonstrate that LOS is an independent risk factor for BPD, with pure LOS showing more pronounced effects on BPD compared to mixed-type sepsis. These results highlight the clinical importance of prioritizing prevention and management of neonatal LOS to reduce BPD risk. Future studies should further refine sepsis classification criteria to enable more precise evaluation of the impact of different types of sepsis on BPD development (Table 3 and Appendix 18).

This meta-analysis confirms that PDA is an independent risk factor for BPD in preterm infants. Stratified analysis by GA showed that in infants with GA < 30 weeks, PDA demonstrated a non-significant trend toward increased BPD risk (OR = 1.72, 95% CI [0.98–3.02]), while in infants with GA ≤ 32 weeks, PDA significantly increased the risk of BPD (OR = 1.74, 95% CI [1.29–2.35]). Although there was no statistically significant difference in effect sizes between the two groups (P = 0.97), both showed substantial heterogeneity (I2 = 92% and 89%, respectively; Table 3 and Appendix 19), which may be related to inconsistencies in PDA management strategies across study centers. Subgroup analysis based on PDA treatment status revealed that the untreated PDA group had significantly higher BPD risk (OR = 2.31, 95% CI [1.44–3.71]) compared to the treated PDA group (OR = 1.40, 95% CI [1.02–1.93]), with the intergroup difference approaching statistical significance (P = 0.09). Notably, substantial heterogeneity was observed in both subgroups (untreated group I2 = 85%, treated group I2 = 94%), and this may be attributable to variations in treatment protocols and medication choices. The findings suggest that standardized PDA management should be emphasized in clinical practice to reduce the risk of BPD, regardless of GA (Table 3 and Appendix 20).

Sensitivity analyses

Sensitivity analyses were performed for different risk factors. The leave-one-out method was used to test the stability of the meta-analysis results.

The meta-analysis results demonstrated robust associations between BPD risk factors (CA, HDP, GA, MV, oxygen administration, blood transfusion, male sex, and PDA) and outcomes, as evidenced by stable effect sizes observed when each study was sequentially omitted under the random-effects model. Moreover, RDS analysis results also demonstrated stability in the fixed-effects model sensitivity analysis. Sensitivity analysis plots for these risk factors are shown in Appendix 21 Fig. A–I.

The leave-one-out sensitivity analysis revealed heterogeneity in effect estimates for specific risk factors: (1) For PROM, excluding Rojas et al. (2012) increased the pooled OR from 1.10 to above 1.5; (2) In SGA analyses, excluding Nakashima et al. (2021) markedly elevated the OR from 1.4 to above 5.5; (3) Regarding sepsis, excluding either Nakashima et al. (2021) or Rojas et al. (2012) increased the OR from 1.10 to above 1.4. While these modifications led to quantitatively important changes, the consistent direction of effect (all ORs > 1) and the maintenance of statistical significance (all 95% CIs excluding one) support the robustness of the primary conclusions. Sensitivity analysis plots for these risk factors are shown in Appendix 21 Fig. J–L.

Publication bias

There were more than 10 original studies focusing on GA as a risk factor for BPD. Therefore, a publication bias test was performed for this factor. The results of this test were presented in a funnel plot and assessed by an Egger’s test. The funnel plot showed an asymmetric pattern (Appendix 22), but the Egger’s test suggested that P = 0.789 > 0.05. This finding indicates no evidence of publication bias.

Discussion

In the present meta-analysis, risk factors for BPD were identified at three distinct periods: during the antenatal, intrapartum, and postpartum phases. The antenatal risk factors encompassed CA, PROM, and HDP. Intrapartum factors included lower GA, male sex, and SGA. Postnatal risk factors included MV, oxygen administration, blood transfusion, PDA, sepsis, and RDS.

Antenatal risk factors

CA is primarily a maternal inflammatory response caused by ascending microorganisms (bacteria, fungi, mycoplasma hominis) that enter the amniotic cavity through the cervix (Kim et al., 2015). CA was diagnosed by clinical findings, including maternal fever, leukocytosis, and uterine tenderness, or the presence of acute inflammatory changes in tissue samples taken from the amnion, chorion, and parietal decidua (Mittendorf et al., 2005). CA may accelerate lung maturation, but it also causes lung inflammation and subsequent lung injury in intubated infants, promoting the development of BPD (Watterberg et al., 1996). CA infection is also the leading cause of preterm birth, and lower gestational ages are associated with higher incidence rates (Kim et al., 2015). Inflammatory stimuli can induce alveolar structural damage; however, the immature lungs of preterm infants possess intrinsic repair and remodeling capacities. When preterm birth occurs, the disruption of ongoing lung development—coupled with sustained inflammation from conditions like CA—impairs these reparative processes, thereby increasing susceptibility to BPD (Hütten & Kramer, 2014). The current study’s finding of a significant association between CA and BPD (OR = 1.52, 95% CI [1.23–1.87]) suggests that CA is a risk factor for BPD. The observed heterogeneity was significantly related to the different stages and grades of CA. A study by Hartling, Liang & Lacaze-Masmonteil, (2012) concluded that CA cannot be definitively considered a risk factor for BPD. Further clinical studies and observations are required to investigate this association more thoroughly.

The diagnosis of PROM was made by a combination of the following methods: checking for leaking amniotic fluid during speculum examination, basic pH testing, and quantitative testing for amniotic proteins (Nakamura et al., 2020). The association of PROM with preterm delivery, sepsis, CA, and pneumonia has been previously documented (Bonasoni et al., 2021; Lorthe, 2018; Lv, Huang & Ma, 2024; Tchirikov et al., 2017), and these complications may contribute to BPD development. The meta-analysis demonstrated a consistent association between PROM and increased BPD risk across individual studies, with the pooled results reaching statistical significance (OR = 1.42, 95% CI [1.02–1.98], P < 0.05). While substantial heterogeneity (I2) was observed, the robustness of this association is supported by three key factors: (1) all included studies independently showed directional consistency in risk elevation, (2) the effect size remained statistically significant despite a limited sample size, and (3) the lower bound of the 95% CI [1.02] exceeded the null threshold. These findings suggest that PROM should be considered a clinically relevant risk factor for BPD, though the magnitude of effect may require refinement through future studies.

The available evidence indicates that the production of pulmonary vessels plays a role in alveolar growth and the maintenance of alveolar structure following birth (Thébaud & Abman, 2007). Maternal factors that disrupt or impede angiogenesis may result in impaired lung vessel development and impaired bronchoalveolar differentiation in the fetus, thereby increasing the incidence of BPD in preterm infants (Grover et al., 2005). This may provide a potential explanation for the observed relationship between HDP and BPD. The results of the current study indicate that HDP is a risk factor for BPD (OR = 2.73, 95% CI [1.31–5.69]). These findings are in alignment with those of Gemmell et al. (2016) and contrasting those of Razak et al. (2018). However, further research is required to substantiate these observations.

Intrapartum risk factors

The findings of this study indicate that GA, SGA, and male sex are risk factors for BPD (MD = −1.86, 95% CI [−2.35 to −1.38]; OR = 3.14, 95% CI [1.03–9.60]; OR = 1.41, 95% CI [1.14–1.75]). It has been demonstrated that preterm infants exhibit a deficiency of lung surface-active substances that can result in RDS (Whitsett, Wert & Weaver, 2015). Additionally, preterm infants are susceptible to inflammation, oxidative stress, and other injuries when their lungs are developing in the tubular or vesicular phase (Lignelli et al., 2019). It has been suggested that there is a sex-dependent response of pulmonary fibroblasts to hyperoxia exposure, with a more pronounced systemic inflammatory response observed in male patients (Balaji et al., 2018; Leroy et al., 2018; Lignelli et al., 2019). This may be a key factor contributing to the observed increased risk of BPD in males. SGA fetuses frequently necessitate prolonged mechanical ventilation and elevated oxygen concentrations due to developmental constraints, thereby elevating their risk of BPD (Bardin, Zelkowitz & Papageorgiou, 1997; De Jesus et al., 2013). Some studies have identified asphyxia and decreased amniotic fluid as intrapartum risk factors for BPD (Lu et al., 2021; Nakamura et al., 2020); however, their reported effect sizes could not be combined due to the limited availability of raw data. Further data are required in the future to assess the intrapartum risk factors for BPD.

Postnatal risk factors

Oxygen therapy and mechanical ventilation are frequently used to sustain blood oxygen levels in preterm infants with hypoxemia (Thébaud et al., 2019). While these measures can provide temporary relief from respiratory distress, they may also result in damage to the delicate lung tissue. Prolonged exposure to elevated levels of oxygen has been demonstrated to induce an increase in free radicals, which in turn has been shown to cause oxidative stress and disrupt the normal metabolic balance between endothelial and epithelial cells in the lung tissue (Thébaud et al., 2019). Mokres et al.’s (2010) study demonstrates that prolonged MV of developing lungs, in the absence of hyperoxia, can impede alveolar septation and angiogenesis, while simultaneously increasing apoptosis and lung elastin. Oxygen administration and MV were shown to be risk factors for BPD in the current study, and this risk increased with longer MV duration. Therefore, it is imperative to reduce oxygen administration and the duration of MV in order to combat BPD.

The administration of a blood transfusion may be associated with an increased risk of oxidative damage, infection, and other complications (Collard, 2006). The results of the current study indicate that blood transfusion is a risk factor for BPD (OR = 1.38, 95% CI [1.06–1.81]). While crude comparisons suggested differences in baseline characteristics (e.g., gestational age, birth weight, and mechanical ventilation duration) between transfused and non-transfused populations, it is important to note that all included original studies conducted multivariate analyses to statistically adjust for these confounding factors. Therefore, the observed association reflects an independent effect of blood transfusion after controlling for these variables.

PDA is significantly associated with an increased risk of BPD (OR = 1.75, 95% CI [1.35–2.27]). These findings are consistent with those of Deng et al. (2022). An increase in blood flow to the pulmonary circulation across the PDA after birth can result in pulmonary edema and prolonged ventilatory support, which may in turn lead to an increased risk of ventilator-induced lung injury (Varsila et al., 1995). An excessive pulmonary blood flow, which results in neutrophil margination and subsequent inflammation, represents a risk factor for BPD (Slaughter et al., 2017). Nevertheless, in clinical practice, conservative treatment for PDA remains the prevailing approach (Letshwiti et al., 2017). Subgroup analysis based on PDA treatment status revealed that the untreated PDA group had significantly higher BPD risk compared to the treated PDA group. These findings suggest that early detection and standardized management of PDA may potentially reduce the incidence of BPD.

It is now well established that inflammation plays a pivotal role in the pathogenesis of BPD (Dankhara et al., 2023). In animal models, it has been demonstrated that infection can trigger a pro-inflammatory response, which in turn results in an interruption in normal lung development. This leads to alveolar simplification and disruption of normal angiogenesis (Franco et al., 2002). Additionally, inflammation can result in dysfunction of the lung endothelium, which gives rise to structural alterations in vascular development. This results in a reduction in the density and number of pulmonary vessels during the process of alveolar development (Choi et al., 2016). The findings from the current study indicate that sepsis is a risk factor for BPD (OR = 1.88, 95% CI [1.44–2.46]). These results are consistent with those of Van Marter et al. (2002). Notably, the subgroup analysis revealed that LOS confers a substantially higher BPD risk compared to EOS/LOS presentations. This differential risk profile may reflect the more vulnerable developmental window of preterm lungs to inflammatory insults during the late postnatal period, when alveolarization is actively progressing. However, the prophylactic use of antibiotics on a widespread basis is not recommended. It has been suggested that early exposure to antibiotics in low-risk early-onset sepsis is a risk factor for BPD (Shi et al., 2024). Improved hand hygiene, aggressive prevention and treatment of nosocomial infections, and rational use of antibiotics are essential.

The results of the current study indicate that RDS is a risk factor for BPD (OR = 6.37, 95% CI [4.0–10.13]). There may be lung surface material deficiency, mechanical ventilation injury, inflammatory response, and oxidative stress in RDS (Carvalho et al., 2018; Marseglia et al., 2019). These processes are intertwined and collectively contribute to the onset and progression of BPD.

To systematically reduce the risk of BPD, a comprehensive perinatal prevention and control system should be established, with particular emphasis on preventing preterm birth whenever possible. During the prenatal phase, proactive measures to prolong gestation should be prioritized. Emphasis should be placed on preventing CA (broad-spectrum antibiotic therapy) and HDP (strict blood pressure control combined with magnesium sulfate for neuroprotection), while cases with PROM require both antibiotics and corticosteroids to promote fetal lung maturation. During delivery, delayed cord clamping should be implemented, and high-risk preterm infants should receive protective ventilation strategies (SpO2 90–94%). In the postnatal phase, early screening and treatment for PDA (prompt pharmacological or surgical intervention for significant hemodynamic changes) are essential, along with optimized respiratory support (minimally invasive surfactant administration and low tidal volume mechanical ventilation), strict prevention of LOS (enhanced infection control measures), prudent blood transfusion criteria, and individualized nutritional support. This integrated approach from pregnancy through postnatal care aims to minimize both the occurrence of prematurity and its pulmonary complications.

Limitations of the study

It should be noted that this study is subject to several limitations. Only data that met the 2001 NICHD diagnostic criteria for BPD were extracted from the original study. Consequently, we excluded findings based on alternative criteria, which led to the omission of some original data. Additionally, the current analysis only included studies published in English, potentially omitting significant research published in other languages. Although many influencing factors were analyzed, many studies addressing the same factor were limited by the inclusion criteria, reducing the statistical power and increasing heterogeneity. The included studies were observational and potentially subject to residual confounding bias (e.g., unmeasured variables may bias the association). While the current study adhered strictly to PRISMA guidelines, we acknowledge that some observational studies containing relevant BPD risk factor data may have been overlooked if they did not include standardized ‘risk factor’ terminology in their titles or abstracts. This inherent constraint of keyword-based searches could potentially affect the comprehensiveness of the evidence synthesis, particularly for studies reporting risk associations without explicit methodological framing. Finally, some variables addressed by fewer studies could not be formally assessed for publication bias due to statistical limitations.

Conclusion

This systematic review and meta-analysis highlights the multifactorial and temporally distributed nature of BPD risk factors, spanning the antenatal (CA, PROM, HDP), intrapartum (lower GA, SGA, male sex), and postnatal (MV, oxygen administration, PDA, sepsis, RDS, blood transfusions) periods. The cumulative and potentially synergistic impact of these factors underscores the need for a phase-specific, multidisciplinary approach to BPD prevention, integrating obstetrical and neonatal care.

While this study identifies key risk factor associations, further research is needed to explore their interactions, epigenetic mechanisms, and individualized risk stratification. Future efforts should focus on dynamic predictive modeling and precision-based interventions to mitigate BPD development. Clinicians must remain vigilant across the perinatal continuum, implementing timely, evidence-based strategies to optimize outcomes for at-risk infants.

Ultimately, reducing BPD burden requires early identification of high-risk pregnancies, proactive management of modifiable factors, and tailored respiratory support to minimize postnatal lung injury and promote healthy pulmonary development.

Supplemental Information

Supplemental Information 1 Search strategy for a systematic review of risk factors associated with bronchopulmonary dysplasia in preterm infants.

Detailed search strategy for the systematic review and meta-analysis on risk factors of bronchopulmonary dysplasia, including databases (e.g., PubMed, Embase), search terms (e.g., ’prematurity’, ’ bronchopulmonary dysplasia ’ ,’risk factors’), and Boolean operators (AND/OR).

Supplemental Information 2 Quality assessment of included studies in the systematic review of bronchopulmonary dysplasia risk factors.

Quality assessment of included studies using the Newcastle-Ottawa Scale (NOS) for cohort studies and case-control studies. The evaluation covers three key domains: (1) selection bias (representativeness of exposed cohort, selection of non-exposed cohort, etc.), (2) comparability (control for confounding factors), and (3) outcome assessment (assessment method and follow-up adequacy). Results are categorized as low, moderate, or high risk of bias.

Supplemental Information 3 Forest plot of CA (random).

Forest plot of chorioamnionitis (CA) as a risk factor for bronchopulmonary dysplasia (BPD) using random-effects model (pooled OR=1.52, 95% CI 1.23-1.87, I²=55%)

Supplemental Information 4 Forest plot of PROM (random).

Forest plot of premature rupture of membranes (PROM) as a risk factor for bronchopulmonary dysplasia (BPD) using random-effects model (pooled OR=1.42, 95% CI 1.02-1.98, I²=63%)

Supplemental Information 5 Forest plot of HDP (random).

Forest plot of hypertensive disorders pregnancy (HDP) as a risk factor for bronchopulmonary dysplasia (BPD) using random-effects model (pooled OR=2.73, 95% CI 1.31-5.69, I²=69%)

Supplemental Information 6 Forest plot of GA (random).

Forest plot of gestational age (GA) as a risk factor for bronchopulmonary dysplasia (BPD) using random-effects model (pooled MD=-1.86, 95% CI -2.35, -1.38, I²=97%)

Supplemental Information 7 Forest plot of sex (random).

Forest plot of sex as a risk factor for bronchopulmonary dysplasia (BPD) using random-effects model (pooled OR=1.41, 95% CI 1.14-1.75, I²=51%)

Supplemental Information 8 Forest plot of SGA (random).

Forest plot of small for gestational age (SGA) as a risk factor for bronchopulmonary dysplasia (BPD) using random-effects model (pooled OR=3.14, 95% CI 1.03-9.60, I²=86%)

Supplemental Information 9 Forest plot of MV (random).

Forest plot of mechanical ventilation ( MV) as a risk factor for bronchopulmonary dysplasia (BPD) using random-effects model (pooled MD=16.55, 95% CI 9.68-23.41, I²=98%)

Supplemental Information 10 Forest plot of oxygen administration (random).

Forest plot of oxygen adnimistration as a risk factor for bronchopulmonary dysplasia (BPD) using random-effects model (pooled MD=50.91, 95% CI 37.40-64.42, I²=97%)

Supplemental Information 11 Forest plot of blood transfusion (random).

Forest plot of blood transfusion as a risk factor for bronchopulmonary dysplasia (BPD) using random-effects model (pooled OR=1.38, 95% CI 1.06-1.81, I²=86%)

Supplemental Information 12 Forest plot of PDA (random).

Forest plot of patent ductus arteriosus (PDA) as a risk factor for bronchopulmonary dysplasia (BPD) using random-effects model (pooled OR=1.75, 95% CI 1.35-2.27, I²=91%)

Supplemental Information 13 Forest plot of sepsis (random).

Forest plot of sepsis as a risk factor for bronchopulmonary dysplasia (BPD) using random-effects model (pooled OR=1.88, 95% CI 1.44-2.46, I²=87%)

Supplemental Information 14 Forest plot of RDS (fixed).

Forest plot of r espiratory distress syndrome (RDS) as a risk factor for bronchopulmonary dysplasia (BPD) using fixed-effects model (pooled OR=6.37, 95% CI 4.0-10.13, I²=0%)

Supplemental Information 15 Random-effects model forest plot for GA and BPD stratified by study design.

Forest plot of gestational age (GA) as a risk factor for bronchopulmonary dysplasia (BPD), stratified by study design using random-effects model. Mean differences (MD) with 95% confidence intervals are presented separately for cohort (MD= -1.82 weeks, 95% CI -2.31,-1.33) and case-control studies (MD= -2.10 weeks, 95% CI -4.84, 0.65). Heterogeneity was quantified by I² statistics (cohort: I²=97%; case-control: I²=99%). Test for subgroup differences: p=0.84.

Supplemental Information 16 GA-Stratified Analysis of Mechanical Ventilation and Bronchopulmonary Dysplasia: A Random-Effects Model Forest Plot.

Different GA stratifications on MV duration through subgroup analysis. In the very preterm infant subgroup (GA <30 weeks), the pooled results demonstrated a significantly prolonged MV duration (MD=21.28 days, 95% CI 5.81-36.76), albeit with extremely high heterogeneity (I²=97%). Similarly, in the very preterm infant subgroup (GA ≤32 weeks), MV duration was also significantly extended (MD=13.80 days, 95% CI 4.82-22.79), with even higher heterogeneity (I²=99%). T he test for subgroup differences showed no statistical significance (P=0.41, I²=0%).

Supplemental Information 17 Sepsis and BPD risk across GA strata: forest plot Meta-analysis.

Random-effects model forest plot of sepsis as a risk factor for BPD, stratified by gestational age groups (<30 weeks, ≤32 weeks). Pooled odds ratios (ORs) with 95% confidence intervals are shown for each stratum. Heterogeneity quantified by I² statistics.Test for subgroup differences: p=0.50.

Supplemental Information 18 Diagnostic criteria-stratified association between neonatal sepsis and Bronchopulmonary Dysplasia: a random-effects model forest plot.

Forest plot demonstrating the association between neonatal sepsis (stratified by diagnostic criteria: late-onset sepsis (LOS) vs LOS / early-onset sepsis (EOS) ) and bronchopulmonary dysplasia (BPD) using random-effects meta-analysis. Pooled odds ratios (ORs) with 95% confidence intervals are displayed for each subgroup. Heterogeneity was assessed using I² statistics. Test for subgroup differences: p=0.04.

Supplemental Information 19 PDA and BPD risk across GA strata: forest plot Meta-analysis.

Random-effects model forest plot of patent ductus arteriosus (PDA) as a risk factor for BPD, stratified by gestational age groups (<30 weeks, ≤32 weeks ). Pooled odds ratios (ORs) with 95% confidence intervals are shown for each stratum. Heterogeneity quantified by I² statistics.Test for subgroup differences: p=0.97.

Supplemental Information 20 PDA and BPD risk across treatment status strata: forest plot Meta-analysis.

Forest plot of PDA-associated BPD risk stratified by treatment exposure (treated [pharmacologic/surgical] vs untreated). Pooled odds ratios (OR) with 95% CIs derived from random-effects models. Heterogeneity: treated (I²=94%), untreated (I²=85%). Subgroup difference p=0.09.

Supplemental Information 21 Sensitivity analysis BPD risk factors.

Sensitivity analysis evaluating the robustness of all identified risk factors for bronchopulmonary dysplasia (BPD). Effect estimates (odds ratios or m ean difference ) were recalculated by sequentially excluding individual studies.

Supplemental Information 22 Funnel plot assessing publication bias for gestational age (GA) as a risk factor for bronchopulmonary dysplasia (BPD).

Each point represents an individual study’s effect size (MD in weeks) against its precision (1/SE). Asymmetry was quantified using Egger’s test (p=0.789).

Supplemental Information 23 Rationale and contribution.

With the development of medical technology, more and more premature babies are surviving, but bronchopulmonary dysplasia (BPD) has a high prevalence in the very premature population, and these children with BPD have a long initial hospital stay, a high frequency of hospital readmissions, and a high percentage of subsequent admissions to the pediatric intensive care unit for lung disease, which may even be secondary to pulmonary arterial hypertension. Treatment of BPD is limited and there are no safe and effective medications. Therefore, prevention is more important, and knowledge of risk factors is the basis for identifying preventive measures. This is why a meta-analysis is needed to summarize the relevant risk factors.

We conducted a comprehensive, evidence-based study on the risk factors for BPD in preterm infants, evaluating a total of 12 potential risk factors, including three different time periods: prenatal, intrapartum and postnatal.

Supplemental Information 24 Checklist.

Additional Information and Declarations

Competing Interests

The authors declare that they have no competing interests.

Author Contributions

Ping Xiong conceived and designed the experiments, performed the experiments, analyzed the data, prepared figures and/or tables, authored or reviewed drafts of the article, and approved the final draft.

Lei Li conceived and designed the experiments, performed the experiments, analyzed the data, prepared figures and/or tables, authored or reviewed drafts of the article, and approved the final draft.

Zhangbin Yu conceived and designed the experiments, performed the experiments, authored or reviewed drafts of the article, and approved the final draft.

Yuanlin Pu conceived and designed the experiments, authored or reviewed drafts of the article, and approved the final draft.

Hong Tang conceived and designed the experiments, authored or reviewed drafts of the article, and approved the final draft.

Data Availability

The following information was supplied regarding data availability:

The raw measurements are available in the Files S3 and Table 1.

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
