# Peer review of "Risk factors for bronchopulmonary dysplasia in preterm infants: a systematic review and meta-analysis"

_PeerJ, doi:10.7717/peerj.20202_

## Round 0.1 · original submission · Major Revisions

The reviewers have provided detailed feedback to aid you in improving your manuscript.

·

Basic reporting

The manuscript submitted by Xiong and colleagues investigates the risk factors for bronchopulmonary dysplasia (BPD) in preterm infants through a systematic review and meta-analysis.
The manuscript is generally written in clear and comprehensible language; however, some sections are overly complex and difficult to understand. Additionally, certain parts consist of mere listings of information without a clear explanation of how they support the study’s findings.

The background or research in the Introduction part is logically structured and provides sufficient information; however, some sections are inadequately described. These points will be addressed in the Additional Comments.

Regarding literature citations, some references are presented in the format of "Our results are consistent with those of **," without providing any details about the cited study. As a result, the necessity and validity of these citations are unclear from the manuscript. It is essential to briefly describe the referenced studies and explain how they differ from or relate to the present study.

Experimental design

The manuscript is original and aligns well with the Aims and Scope of the journal. The Research Question is well-defined, meaningful, and relevant to the field. This study conducts a systematic review, adhering to the PRISMA guidelines, which serve as the standard framework for reporting systematic reviews, ensuring methodological rigor and a high level of quality. Objectivity is maintained by conducting literature searches and data extraction independently by two researchers. Furthermore, data synthesis is performed using results adjusted through multifactor analysis, effectively addressing potential confounding factors.

In this systematic review, the inclusion criteria specify "3. The focus of the study is on risk factors or influencing factors for BPD." The search strategy relies on MeSH terms and free-text terms related to "risk factors." However, many observational studies, including case-control and cohort studies, investigate potential risk factors, even if not explicitly stated in the title or abstract. Thus, some observational studies containing relevant BPD risk factor data may not include the specified MeSH or free-text terms, and any study comparing BPD and non-BPD groups with patient background data could qualify as eligible. While the large number of potential studies may require selection, it raises concerns about whether a comprehensive search addressing the clinical question has been appropriately conducted. For example, the following study includes data on HDP but is not included in this systematic review (https://pubmed.ncbi.nlm.nih.gov/30770862/).

In the Methods section, some definitions that should have been pre-specified before initiating the systematic review are missing. It remains unclear whether these definitionss were predetermined but not reported or were not defined beforehand. These issues will be further addressed in the Additional Comments section.

Given the advancements in pathological evaluation and intervention/treatment methods for BPD, it is reasonable to assume that BPD risk factors may change over time. However, the inclusion criteria do not specify the publication period of the studies included in the search, nor does the search strategy mention any publication year restrictions. While many of the included studies appear to be relatively recent, it is unclear whether this reflects a comprehensive search across all publication years or an implicit time restriction. For this type of study, it is essential to explicitly define the time period for included publications in the search strategy.

Validity of the findings

The search strategy used for the literature search and the flowchart of the literature search are properly presented in the manuscript and supplementary materials.

The conclusion clearly summarizes the key findings from the systematic review and meta-analysis, stating that BPD risk factors are multifaceted and occur across different perinatal periods.

Additional comments

Line73. The incidence of BPD is more strongly influenced by gestational age than by differences between centers and regions. Therefore, prevalence data without information on the study population is not meaningful. If prevalence is to be reported, it is necessary to define the target population, such as extremely preterm infants (gestational age <28 weeks) or VLBW infants.

Line75. Preterm infants often require oxygen therapy even if they do not develop BPD. Therefore, when describing the clinical characteristics or definition of BPD, it is necessary to specify the prolonged need for oxygen therapy, such as beyond 28 days of life or up to 36 weeks postmenstrual age.

Line 89- Methods
The definitions of chorioamnionitis, mechanical ventilation, oxygen exposure, patent ductus arteriosus, and sepsis should be established before screening the studies identified in the literature search. Although some definitions are mentioned in the Discussion, they are cited from sources outside the included studies, making it unclear whether these criteria were uniformly applied when extracting outcomes. These definitions must be explicitly stated in the Methods section.

It is also necessary to clarify whether mechanical ventilation refers only to invasive management, what specific type of oxygen exposure is considered, and how patent ductus arteriosus is defined—whether it is diagnosed via ultrasound, or if it includes only cases requiring pharmacological treatment or surgery.

Line120. BDP is a typo for BPD.

Line 291-293 The intent of this sentence is not clear.

Line 304-306.
Even with a limited sample size, each individual study independently shows that PROM increases the risk of BPD, and the combined results also indicate a statistically significant association with PROM at the 0.05 significance level. Therefore, despite the high I² value, PROM can still be considered a risk factor for BPD. Given this, it is reasonable to expect that adding further studies would not significantly alter the overall conclusion.

Line 349-352 Didn't the meta-analysis use data adjusted through multivariable analysis for the factors mentioned here?

·

Basic reporting

This meta-analysis aims to identify the risk factors for bronchopulmonary dysplasia (BPD) in preterm infants born before 32 weeks of gestation (WG) and with a birth weight less than 1500g. The article is clearly written, and the study follows established reporting standards. The PRISMA checklist is thoroughly detailed in the appendix, providing transparency in how the study adhered to recommended guidelines. Additionally, the methodology for the search and inclusion of articles is well-documented, enhancing reproducibility. The studies included were evaluated using the Newcastle-Ottawa Scale (NOS) and are of generally good quality, which is a strong point for the validity of the findings.

Regarding the definition of BPD, it would be helpful to explicitly include the definition in the methods section, alongside the cited source, for clarity. This would ensure that all readers have a uniform understanding of the condition under study.

A minor remark: for Figure S14 (Egger’s test of MV), it would be beneficial to place the figure on a white background to improve readability and visual clarity.

Experimental design

The design of the study is generally sound and follows appropriate methodology for a meta-analysis. Several strengths of the study should be highlighted:

Comprehensive Search Strategy: The search methodology is detailed, ensuring a comprehensive review of the relevant literature. This approach adds rigor to the study by including a broad range of studies.

Quality Assessment: The authors have assessed the included studies using a well-recognized scale (NOS), which ensures that the studies are of high quality and that bias is minimized at the study selection stage.

Heterogeneity Testing: The study performs heterogeneity tests for each variable, which is a necessary step in meta-analysis to understand the variability across studies. However, the observed heterogeneity, while acknowledged, was not deeply explored or explained. A more detailed investigation of potential sources of heterogeneity would provide further insight into the findings.

Publication Bias: The authors assess publication bias for the variable studied in the largest number of studies, which is a commendable approach. However, further details on how publication bias was assessed for other variables would add to the transparency and rigor of the methodology.

Data Presentation: The study uses aggregated data from case-control and cohort studies, which is appropriate for identifying general trends but limits the ability to adjust for important confounders. This is a common challenge in meta-analyses using aggregated data.

Validity of the findings

Heterogeneity and Confounding Factors: There is considerable heterogeneity in most of the variables analyzed, which is expected in a meta-analysis of studies with different designs and populations. However, the impact of this heterogeneity was not fully explored. It would strengthen the interpretation of the findings if the authors could provide further explanation of the sources of this heterogeneity and conduct additional subgroup analyses if possible.

P-values and Statistical Reporting: The authors report p-values, but it would be preferable to provide more specific p-values rather than simply indicating "p < 0.05". For example, stating p = 0.03 would provide more context and improve the precision of the results.

Sex vs. Gender: It is important to replace the term "gender" with "sex" when referring to biological characteristics. This distinction is crucial, as the analysis concerns biological differences, not gender identity (e.g., line 323: change "gender dependent response" to "sex dependent response").

Confounding Bias: The discussion mentions confounding biases but lacks specific details about how these were addressed. One major potential confounder, gestational age, should be more thoroughly discussed. If the authors adjusted for this factor or other potential biases, they should provide details on these adjustments. This would help in understanding the robustness of the findings and their generalizability.

Non-Innovative Findings: While the meta-analysis provides useful data, the findings do not appear to offer particularly novel insights into the risk factors for BPD, as these factors are already well-known in the literature. The authors could provide a more detailed discussion of how their findings add to existing knowledge or suggest future research directions that could be explored to further advance the field.

Methodological Limitations: The meta-analysis relies on aggregated data from case-control and cohort studies, which inherently limits the ability to perform more refined adjustments for confounding factors. This is an important limitation, and the authors should acknowledge it more explicitly in the discussion. More detailed exploration of the biases in the included studies would aid in the interpretation of the findings.

Practical Recommendations: The discussion would benefit from concrete suggestions for practice or strategies to mitigate the identified risk factors for BPD. While providing practical recommendations based on a meta-analysis is not always feasible, it would be valuable to discuss potential pathways for preventing or managing BPD in preterm infants more explicitly.

Additional comments

Conclusion
In conclusion, this meta-analysis is well-conducted, providing valuable insights into the risk factors for bronchopulmonary dysplasia in preterm infants. The methodology is generally sound, with a detailed reporting of the search strategy, quality assessment of included studies, and heterogeneity analysis. However, the study has inherent limitations, especially in terms of confounding factors that could not be fully adjusted for due to the use of aggregated data. While the findings contribute useful knowledge, they are not particularly novel, as the risk factors for BPD are already well-established in the literature. The discussion would benefit from a more detailed exploration of confounding biases, and the authors could consider providing practical recommendations or future research directions.

---

## Round 0.2 · Minor Revisions

**Language Note:** When you prepare your next revision, please either (i) have a colleague who is proficient in English and familiar with the subject matter review your manuscript, or (ii) contact a professional editing service to review your manuscript. PeerJ can provide language editing services - you can contact us at [email protected] for pricing (be sure to provide your manuscript number and title). – PeerJ Staff

·

Basic reporting

The manuscript submitted by Xiong and colleagues investigates the risk factors for bronchopulmonary dysplasia (BPD) in preterm infants through a systematic review and meta-analysis. The manuscript is generally written in clear and comprehensible language. Clear, unambiguous, and professional English is used throughout the text.

The background and rationale in the Introduction section are logically structured and provide sufficient context for the study. The literature cited is well-referenced and relevant to the research question. The manuscript's structure conforms to PeerJ standards and disciplinary norms, and in some areas, has been adjusted to enhance clarity.

Figures included in the manuscript are relevant, of high quality, and appropriately labeled and described.

Experimental design

The manuscript is original and aligns well with the Aims and Scope of the journal. The research question is clearly defined, meaningful, and highly relevant to the field. This study conducts a systematic review in accordance with the PRISMA guidelines, ensuring methodological rigor and a high level of quality. Literature search and data extraction were independently conducted by two researchers, maintaining objectivity. Furthermore, data synthesis was performed using results adjusted through multivariable analysis, effectively addressing potential confounding factors.

In their response to the initial review, the authors stated: “We fully acknowledge that some studies investigating BPD risk factors may contain relevant data even without explicit terminology in their titles or abstracts.” Although the search criteria were not revised upon resubmission—which is understandable—the possibility remains that some observational studies containing relevant data on BPD risk factors were not captured, as they did not include terms related to "risk factors."
Therefore, it cannot be concluded that the literature search in this systematic review was fully comprehensive, and this issue should be acknowledged in the manuscript as a limitation of the study.

Validity of the findings

The search strategy used for the literature search and the flowchart of the literature search are properly presented in the manuscript and supplementary materials.

The conclusion clearly summarizes the key findings from the systematic review and meta-analysis, stating that BPD risk factors are multifaceted and occur across different perinatal periods.

---

## Round 0.3 · accepted · Accept

Thank you for addressing the remaining concerns of reviewer 1. The manuscript is now ready for publication.